# Recovery from spindle checkpoint-mediated arrest requires a novel Dnt1-dependent APC/C activation mechanism

Shuang Bai ◉[☉], Li Sun[☉], Xi Wang, Shuang-min Wang, Zhou-qing Luo ◉*, Yamei Wang*, Quan-wen Jin ◉*

State Key Laboratory of Cellular Stress Biology, School of Life Sciences, Faculty of Medicine and Life Sciences, Xiamen University, Xiamen, Fujian, China

☉ These authors contributed equally to this work.
* luozq@xmu.edu.cn (ZL); wangyamei@xmu.edu.cn (YW); jinquanwen@xmu.edu.cn (QJ)

**Data Availability Statement:** All relevant data are within the manuscript and its Supporting Information files.

## Abstract

The activated spindle assembly checkpoint (SAC) potently inhibits the anaphase-promoting complex/cyclosome (APC/C) to ensure accurate chromosome segregation at anaphase. Early studies have recognized that the SAC should be silenced within minutes to enable rapid APC/C activation and synchronous segregation of chromosomes once all kinetochores are properly attached, but the underlying silencers are still being elucidated. Here, we report that the timely silencing of SAC in fission yeast requires $dnt1^+$, which causes severe thiabendazole (TBZ) sensitivity and increased rate of lagging chromosomes when deleted. The absence of Dnt1 results in prolonged inhibitory binding of mitotic checkpoint complex (MCC) to APC/C and attenuated protein levels of Slp1$^{Cdc20}$, consequently slows the degradation of cyclin B and securin, and eventually delays anaphase entry in cells released from SAC activation. Interestingly, Dnt1 physically associates with APC/C upon SAC activation. We propose that this association may fend off excessive and prolonged MCC binding to APC/C and help to maintain Slp1$^{Cdc20}$ stability. This may allow a subset of APC/C to retain activity, which ensures rapid anaphase onset and mitotic exit once SAC is inactivated. Therefore, our study uncovered a new player in dictating the timing and efficacy of APC/C activation, which is actively required for maintaining cell viability upon recovery from the inhibition of APC/C by spindle checkpoint.

## Author summary

During eukaryotic mitotic cycle, the anaphase-promoting complex/cyclosome (APC/C) ubiquitin ligase is tightly regulated to ensure programmed ubiquitination and proteolysis of securin and cyclin B to consequently initiate anaphase entry and mitotic exit. Multifaceted negative regulation of the APC/C activity is mediated primarily through direct binding by its potent inhibitor, the mitotic checkpoint complex (MCC). Here in this study, we have identified an unexpected role for the fission yeast nucleolar protein Dnt1 in maintaining a sufficient level of APC/C co-activator, Slp1$^{Cdc20}$, for anaphase entry, and

**Funding:** This work was supported by grants from the National Natural Science Foundation of China (No. 32170731, No. 31871362, No. 31171298, and No. 31671411) to Q-wJ. The funders had no role in study design, data collection and analysis, decision to publish, or preparation of the manuscript.

**Competing interests:** The authors have declared that no competing interests exist.

restraining overly inhibitory action of MCC on APC/C activation. Interestingly, Dnt1 physically interacts with APC/C upon spindle assembly checkpoint (SAC) activation, this possibly allows it to actively fine tune the association of MCC with APC/C, especially during the process of SAC inactivation and mitotic exit. Our study also revealed that Dnt1 is actively involved in maintaining cell viability when yeast cells are released from the inhibition of APC/C by spindle checkpoint.

## Introduction

The eukaryotic cell cycle comprises two major alternate events, chromosome duplication during interphase and subsequent chromosome segregation during mitosis, usually with extremely high fidelity. When errors do occur, they can have catastrophic consequences, including cell death or genome instabilities. In late mitosis, chromosome segregation and anaphase onset are initiated through the action of a 1.2 MDa multi-subunit E3 ubiquitin ligase known as the anaphase-promoting complex or cyclosome (APC/C) [1]. Most of the APC/C subunits are essential for viability and are conserved in all eukaryotes from yeast to humans [2,3]. To fulfill its proper functions during mitosis, APC/C needs to cooperate with at least two ubiquitin-conjugating (E2) enzymes and one essential co-activator, Cdc20 (Slp1 in the fission yeast *Schizosaccharomyces pombe*), to recruit and ubiquitylate substrates for proteasomal degradation [4–8]. Securin (Cut2 in fission yeast) and cyclin B (Cdc13 in fission yeast) are two major APC/C substrates, and their polyubiquitylation and degradation are critical for anaphase onset and chromosome segregation [9]. In humans, deregulation of these control mechanisms and altered activity of the APC/C can lead to severe mitotic defects and genome instabilities and has been associated with the development of various human cancer types [10–14].

The spindle assembly checkpoint (SAC) is an intricate surveillance mechanism that prolongs mitosis until all chromosomes achieve correct bipolar attachments to spindle microtubules. The core components of the SAC, including Mad1, Mad2, BubR1 (also known as Mad3 in yeasts and worms), Bub1, Bub3 and Mps1 (Mph1 in fission yeast), accumulate on unattached kinetochores and start a signaling cascade that ultimately inhibits Cdc20 [4,15,16]. The mitotic checkpoint complex (MCC) is composed of 3–4 core proteins (Cdc20-BubR1/Mad3-Mad2 with or without Bub3 depending on species) and has been found to be the most potent inhibitor of the APC/C prior to anaphase [4,17,18]. Recent biochemical and structural studies revealed that the human and fission yeast MCCs bind two Cdc20 molecules, one (Cdc20$^{MCC}$) through cooperative binding to Mad2 and Mad3/BubR1 (forming the "core MCC") and the other (Cdc20$^{APC/C}$) through additional binding motifs in BubR1/Mad3 [19–23].

Although it has been well established that the production of MCC and SAC signaling must be inactivated once chromosomes are properly aligned on the spindle before anaphase onset, the molecular mechanisms of SAC inactivation remain obscure [4,17]. So far, about six proteins including p31$^{comet}$, the AAA$^+$ ATPase TRIP13, UbcH10, Spindly, CUEDC2 and phosphatase PP2A have been suggested to be involved in checkpoint inactivation in mammalian cells [24–31]. p31$^{comet}$, TRIP13, CUEDC2 and Spindly ensure timely spindle checkpoint silencing subsequent to kinetochore attachment through promoting the release of Mad2 from MCC or stripping of Mad2 from the kinetochore, respectively [27,28,32–38], while PP2A inactivates SAC by reversing the Mps1-mediated Knl1 phosphorylation necessary for Bub1/BubR1 recruitment [29]. By contrast, the spindle checkpoint-silencing mechanisms in fungi remains

less well explored, limiting the full understanding about this key cellular process. In both budding and fission yeast, type 1 phosphatase PP1 (Dis2 in fission yeast) is required for spindle checkpoint silencing by opposing Aurora B kinase, although their targeted substrates still remain unidentified [39–41]. Whether other proteins and mechanisms are involved in SAC silencing in yeast is currently unclear.

Fission yeast Dnt1 accumulates mainly in the nucleolus throughout the entire cell cycle and was originally identified in a genetic screen for suppressors of the cytokinesis checkpoint defects in the septation initiation network (SIN) mutants [42]. This suppression was later attributed to a mechanism that is mediated by the Cdk1 regulator Wee1 kinase [43]. Dnt1 also appears as a negative regulator of Dma1 in early mitosis [44]. Dma1, the checkpoint protein and E3 ubiquitin ligase, is a distinct spindle checkpoint protein and plays an important role in delaying cytokinesis by inhibiting the SIN when chromosomes are not attached to the mitotic spindle [45–47]. In this study, we identified Dnt1 as a novel factor involved in inhibiting excessive and prolonged MCC-APC/C association and promoting Slp1$^{Cdc20}$ stability, and thus it ensures timely spindle checkpoint inactivation, which is essentially required for maintaining cell viability upon recovery from the inhibition of APC/C by checkpoint.

## Results

### Dnt1 is involved in facilitating proper mitotic chromosome segregation

In the course of our characterizing Dnt1 as a negative regulator of Dma1 in early mitosis [44], we noticed that *dnt1Δ* cells were extremely sensitive to microtubule-destabilizing drug thiabendazole (TBZ), and deletion of *dma1*$^+$ in these cells only slightly reversed this phenotype (Fig 1A). Since the disruption of genes participating in chromosome segregation, such as the kinetochore- or spindle checkpoint-related genes, usually results in TBZ sensitivity [48,49], we assumed that *dnt1*$^+$ gene might have *dma1*$^+$-irrelevant function that is essential for chromosome segregation.

Consistent with this assumption, we found that *dnt1Δ* cells lost minichromosomes (Ch16, *ade6-M216*) at an elevated rate that is almost 100 times higher than that of the wild-type (Fig 1B), and displayed increased frequency of lagging chromosomes and chromosome mis-segregation at mitotic anaphase (Fig 1C). By following the kinetochore separation and spindle dynamics during mitosis under time-lapse microscopy, we found that *dnt1Δ* cells spent almost the same length of time at late prometaphase/metaphase and anaphase A as wild-type cells except for the occasionally observed lagging chromosomes (2 cases in 14 *dnt1Δ* cells but 0 in 17 wild-type cells) at early anaphase, but *dnt1Δ* cells stayed for extended length of time at anaphase B (Fig 1D–1F). These data strongly suggested that Dnt1 is involved in facilitating proper chromosome segregation.

Chromosome segregation is a precisely regulated process involving many proteins, including kinetochore proteins, monopolins, cohesins, chromosome passenger proteins, centromeric heterochromatin proteins, microtubule-binding proteins, and regulators of kinetochore-microtubule attachment [50]. In order to dissect the mechanism behind how Dnt1 is involved in maintaining the fidelity of chromosome segregation, we systematically tested the genetic interactions between *dnt1Δ* and some representative mutants that have been previously reported to cause chromosome missegregation. Strikingly, *dnt1Δ* showed mild to strong negative genetic interactions with almost all these mutants (S1 Fig), indicating that Dnt1 became essential when any of these proteins was absent or mutated. Thus, Dnt1 should be involved in faithful mitotic chromosome segregation in a previously unrecognized manner.

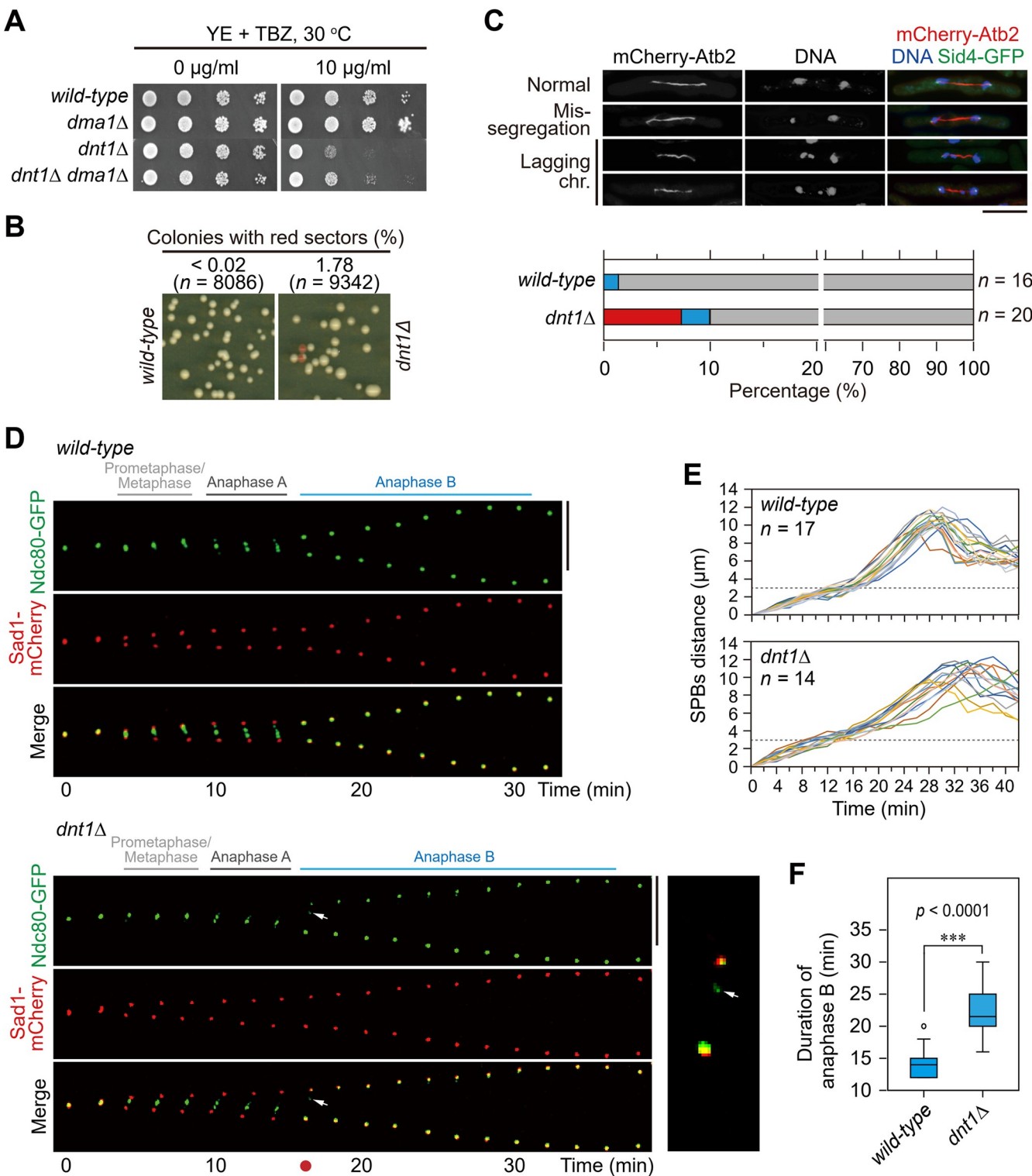

**Fig 1. Characterization on the involvement of Dnt1 in facilitating proper mitotic chromosome segregation. (A)** Tenfold serial dilution analyses of the indicated yeast strains grown on the indicated media to measure the TBZ sensitivity. (**B**) Minichromosome loss rate per division was measured in strains bearing the Ch16 (*ade6-M216*) minichromosome and *ade6-M210* allele (*n* > 8,000). (**C**) (*Left*) Chromosome segregation was observed in anaphase in the indicated cells carrying mCherry-tagged α-tubulin (*atb2*[+]) (red) after being fixed and stained with DAPI (blue). (*Right*) Quantitative analysis of chromosome missegregation and lagging chromosome phenotypes. *n*, numbers of anaphase cells analyzed. Scale bar, 5 μm. (**D-F**) Anaphase B is slightly delayed in *dnt1*Δ cells. (**D**) Time-lapse microscopy of a wild-type and a *dnt1*Δ cell carrying *ndc80-GFP* and *sad1-mCherry* during mitosis. Images were acquired at 2-min

intervals. Arrows indicate a lagging chromosome. The enlarged image corresponding to the time frame indicated by red dot is shown in inset. Scale bar, 5 μm. (**E**) Distance between SPBs (SPB to SPB, marked by Sad1-mCherry) was measured at 2-min intervals. Each line represents data collected from an individual cell. Dashed lines indicate 3 μm, which roughly marks the timing for start of anaphase B in most cases. (**F**) Box-and-whiskers representation of anaphase B (during which pole-to-pole distance increases with separated sister chromatids) duration, in which boxes indicates median and upper and lower quartile and whiskers indicates range of data. The data were extracted and quantified from spindle dynamics measurements in (D). ****, $p < 0.0001$.

## Dnt1 is dispensable for activating the SAC in the absence of kinetochore-microtubule attachment or tension

The spindle checkpoint is another safeguard mechanism that ensures proper chromosome segregation, which is rapidly activated when the kinetochore-spindle microtubule attachment or the tension generated by this attachment is absent or compromised [16]. We then examined whether the SAC is properly activated in *dnt1Δ* cells when the kinetochore-spindle microtubule attachment or the tension was compromised by either the cold-sensitive β-tubulin mutation *nda3-KM311* [51] or the temperature-sensitive cohesin subunit mutation *psc3-1T* [52,53], respectively. In these two experimental set-ups, the accumulation of two major APC/C substrates cyclin B (Cdc13-GFP) and securin (Cut2-GFP) at SPBs or within nuclei, respectively, served as an indicator of SAC activation and mitotic arrest (S2 Fig). Our quantified data showed that the Cdc13-GFP or Cut2-GFP were accumulated at SPBs or within nuclei in *nda3-KM311* or *pst3-1T* cells, respectively, with indistinguishable rate in the presence and absence of *dnt1+* (S2B and S2D Fig), indicating that Dnt1 is largely dispensable for activating SAC in the absence of either attachment or tension.

## Dnt1 facilitates efficient anaphase initiation upon SAC inactivation

The foregoing results rendered us to hypothesize that Dnt1 might have a role in turning off the spindle checkpoint in mitosis. To explore this, we adopted two well-established SAC silencing assays [39,54]. In these assays, the SAC was first robustly activated by the *nda3-KM311* mutant simultaneously carrying *ark1-as3* and then inactivated by addition of ATP analogue 1-NM-PP1 or simply by shifting mitotically arrested cells back to permissive temperature (30˚C) as previously described (Figs 2A and S3A). Because Cdc13 (cyclin B) and Cut2 (securin) localize to the spindle pole bodies (SPBs) or nucleus, respectively, in early mitosis and should be degraded by APC/C to promote metaphase-anaphase transition, the disappearance rate of Cdc13-GFP spot or nuclear Cut2-GFP after 1-NM-PP1 addition or shifting back to 30˚C reflects the SAC inactivation efficacy (Figs 2B and S4A). We found that *dnt1Δ* cells retained high amounts of SPB-localized Cdc13-GFP and nuclear Cut2-GFP for much prolonged period compared to wild-type cells, almost to the same degree as previously identified SAC-inactivation defective mutant *dis2Δ* [39] (Figs 2C, S3B, and S4B). Also, the total cellular protein of Cdc13 or Cut2 was indeed degraded much slower in *dnt1Δ* cells than in wild-type cells recovered from *nda3*-mediated checkpoint arrest (Figs 2D, 2E and S4B). All above data suggested that Dnt1 is required for the timely inactivation of SAC to efficiently initiate anaphase.

## Dnt1 is required for timely dissociation of MCC from APC/C during SAC inactivation

In fission yeast, the key SAC components Mad2 and Mad3 and one molecule of Slp1$^{Cdc20}$ form mitotic checkpoint complex (MCC) which binds to APC/C through another molecule of Slp1$^{Cdc20}$ upon checkpoint arrest, and the recovery from mitotic arrest accompanies the loss of MCC-APC/C binding [22,39,55,56]. To examine whether delayed anaphase initiation upon recovery from mitotic arrest in *dnt1Δ* cells was due to persistent MCC-APC/C binding, we

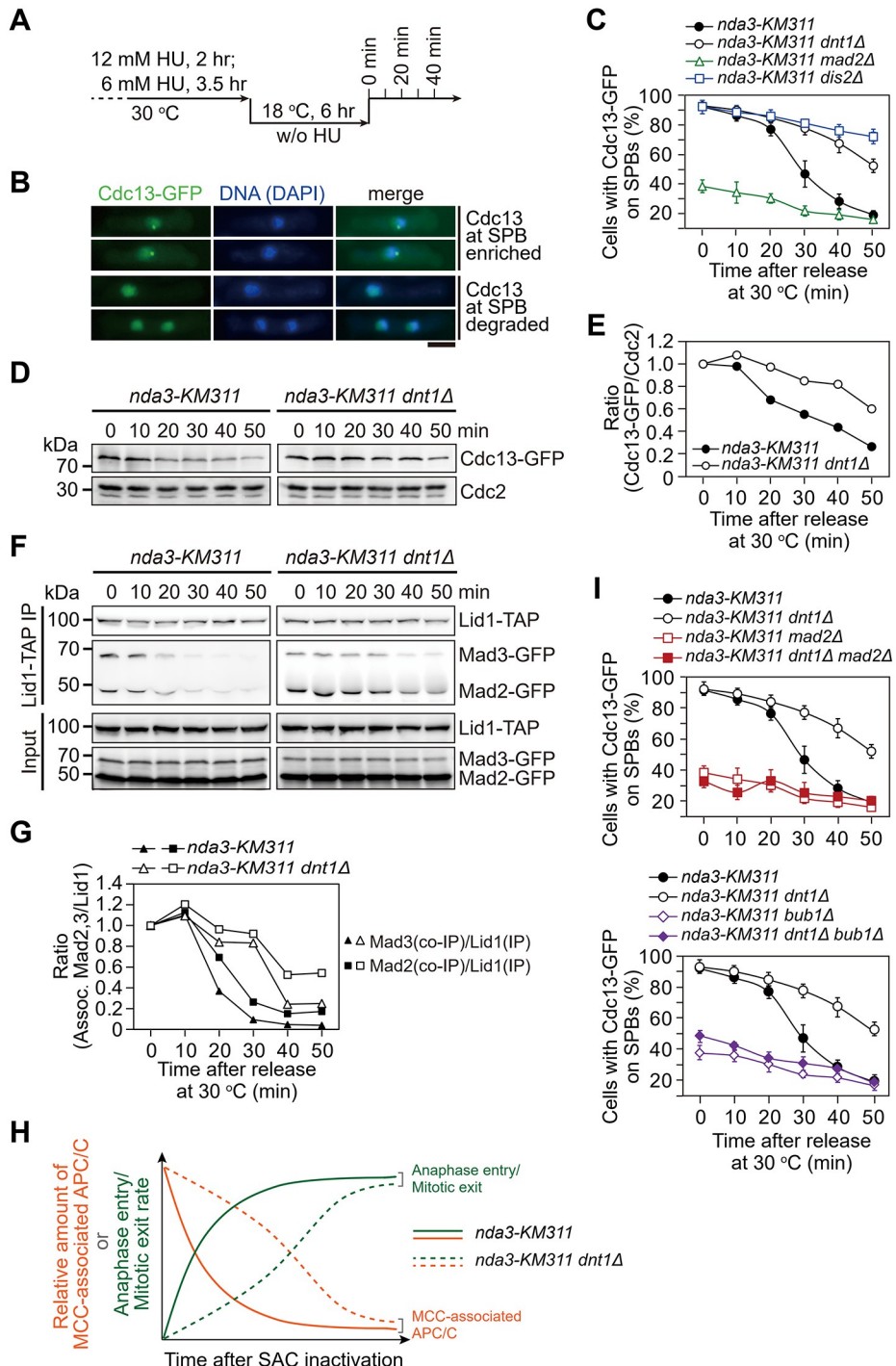

**Fig 2. Dnt1 facilitates timely degradation of Cyclin B and dissociation of Mad2 and Mad3 from APC/C complex upon SAC inactivation.** (**A**) Schematic depiction of the experiment design for (**B-G** and **I**). (**B-E**) Dnt1 is required for timely degradation of Cyclin B upon SAC inactivation. Example pictures of cells with Cdc13-GFP signals enriched or disappeared at spindle pole bodies (SPBs) are shown in (**B**). The percentage of cells with Cdc13-GFP on SPBs was assessed at each time point after shift to 30°C (**C**), which served as an indicator for mitotic exit. Total protein levels of Cdc13-GFP were detected by Western blotting (**D**) and normalized to those of total Cdc2 at each time point, with the relative ratio between Cdc13-GFP and Cdc2 at 0 min set as 1.0 (**E**). Scale bar, 5 μm. (**F-G**) Dissociation of the Mad2

and Mad3 checkpoint proteins from the APC/C during mitotic arrest-and-release is delayed in *dnt1*Δ cells. The association of Mad2 and Mad3 to Apc4/Lid1 was assessed by immunoprecipitation of Apc4/Lid1-TAP and Western blot (**F**). The amount of co-immunoprecipitated Mad2 and Mad3 was normalized to those of total immunoprecipitated Apc4/Lid1 at each time point, with the relative ratio between Mad2-GFP or Mad3-GFP and Apc4/Lid1-TAP at 0 min set as 1.0 (**G**). (**H**) Schematic summary of results shown in C-G. Prolonged MCC-APC/C association in *dnt1*Δ cells anti-correlates with decelerated anaphase entry and mitotic exit upon spindle checkpoint inactivation. (**I**) Depletion of SAC signaling by deletions of *mad2*⁺ or *bub1*⁺ abrogates the delayed SAC inactivation after release from checkpoint arrest in *dnt1*Δ cells. Cells of indicated strains bearing Cdc13-GFP were treated and assessed as in (**B and C**). The experiments were repeated two (**D-G**) or three (**C and I**) times. Error bars correspond to standard deviation (SD).

analyzed the rate of dissociation of the MCC from the APC/C by immunoprecipitations of the APC/C subunit Apc4/Lid1 from mitosis-exiting cells. Indeed, the levels of MCC bound to the APC/C in *dnt1*Δ cells stayed high for longer period than in wild-type cells (Fig 2F and 2G), suggesting that Dnt1 functions for timely and efficient MCC-APC/C dissociation and spindle checkpoint inactivation (Fig 2H), which is required for activating APC/C to degrade Cdc13 and Cut2. Deletions of *mad2*⁺ or *bub1*⁺ significantly lowered the percentage of *dnt1*Δ cells with Cdc13 at SPBs upon *nda3*-mediated checkpoint activation (Fig 2I), indicating that sustained SAC activation and consequent prometaphase- or metaphase-arrest in the absence of Dnt1 relies on prior SAC activation.

## Dnt1 positively regulates the protein levels of Slp1$^{Cdc20}$

Given that MCC functions as a potent inhibitor of APC/C upon spindle checkpoint activation and Dnt1 is required to promote timely dissociation of MCC from APC/C, we were suspicious that one direct consequence of the prolonged MCC-APC/C association in *dnt1*Δ cells might be the inhibition of APC/C activity. Consistent with this assumption, we found that *dnt1*Δ enhanced the growth defects of some temperature-sensitive mutants of essential APC/C subunits, such as *nuc2-663*, *cut9-234*, *cut20-100* and *cut23-547*, though *dnt1*Δ did not have any negative genetic interactions with two other APC/C mutants *apc15*Δ and *slp1-mr63*, both are defective in spindle checkpoint arrest [55, 57] (Fig 3A). Most strikingly, we observed that *dnt1*Δ rescued *slp1-362* surprisingly well (Fig 3A). This specific genetic interaction suggested a possible role of Dnt1 in MCC-APC/C dissociation through regulating Slp1$^{Cdc20}$. To test this hypothesis, we examined the protein levels of the full-length Slp1$^{Cdc20}$ in *nda3-KM311 dnt1*⁺ and *nda3-KM311 dnt1*Δ cells after being released from metaphase-arrest. Surprisingly, Slp1$^{Cdc20}$ was slightly, but appreciably and reproducibly, less abundant (ranging from roughly 20% to 50% at different time points) in *dnt1*Δ cells than in wild-type cells (Fig 3B), suggesting Dnt1 may indeed positively regulate the levels of intact Slp1$^{Cdc20}$. In addition, this regulation of Slp1$^{Cdc20}$ stability by Dnt1 is Dma1-independent, as the *dnt1*Δ *dma1*Δ double mutant has a similar level and degradation profile of Slp1$^{Cdc20}$ as *dnt1*Δ single mutant (S5 Fig).

## Artificially increased Slp1 abundance mitigates TBZ sensitivity of *dnt1*Δ cells

Based on the above results, we wondered whether the decreased levels of Slp1$^{Cdc20}$ may be the major cause of the TBZ sensitivity and the defective anaphase initiation upon SAC inactivation in *dnt1*Δ cells. To test this possibility, we artificially increased Slp1$^{Cdc20}$ abundance by expressing one or two extra copies of *slp1*⁺ under its endogenous regulatory sequences (Fig 3C). Interestingly, two and three copies of *slp1*⁺ restored abundance of Slp1$^{Cdc20}$ in metaphase-arrested *dnt1*Δ cells to close to or slightly above the endogenous level in wild-type cells (Fig 3D).

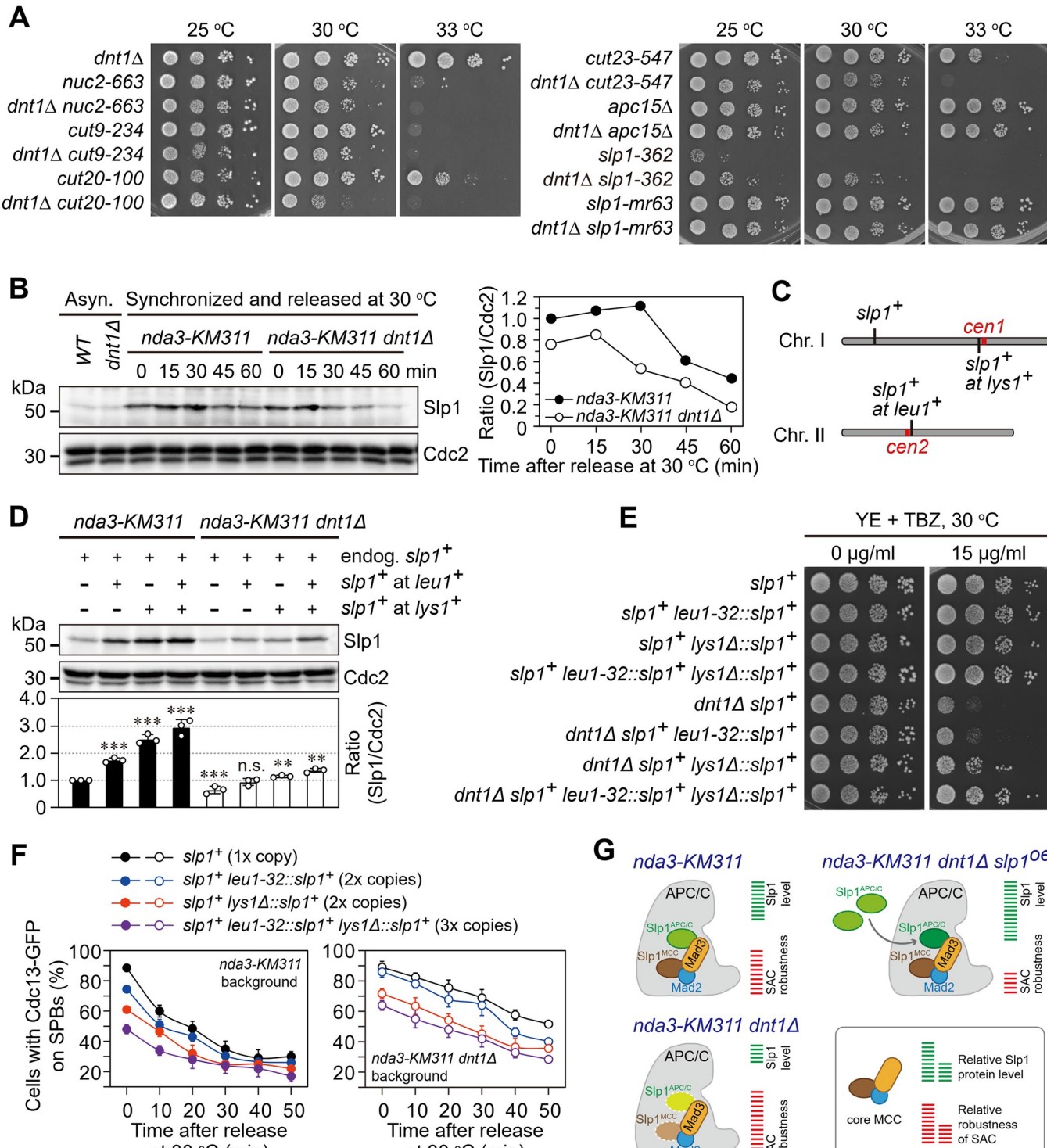

**Fig 3. Dnt1 is required for maintaining the protein levels of Slp1^Cdc20 upon SAC activation.** (**A**) *dnt1Δ* rescues the temperature-sensitivity of *slp1-362* but not other loss-of-function APC/C mutants. Serial dilutions (10-fold) of the indicated strains were spotted on YE plates and incubated at the indicated temperatures. Note that temperature-sensitivity of most loss-of-function APC/C mutants is exacerbated by *dnt1Δ*, but *dnt1Δ* rescues *slp1-362*. In addition, *dnt1Δ* does not have any negative genetic interactions with two other APC/C mutants, *apc15Δ* and *slp1-mr63*, both are defective in spindle checkpoint arrest. (**B**) Slp1^Cdc20 levels are slightly reduced in *dnt1Δ* cells compared to wild-type cells during metaphase arrest and SAC inactivation. Strains with indicated genotypes were grown, treated and synchronized as in Fig 2A. Mid-log phase samples at 30°C were also collected as asynchronous cultures (Asyn.). (*Left*) Samples were subjected to Western blot analyses using anti-Slp1 and anti-Cdc2 antibodies. (*Right*) Slp1^Cdc20 levels were normalized to those of total Cdc2 at

each time point, with the relative ratio between Slp1$^{Cdc20}$ and Cdc2 at time 0 min in *nda3-KM311* set as 1.0. (**C**) Schematic depiction of the genomic positions of endogenous *slp1$^+$* locus and ectopic *slp1$^+$* at *lys1$^+$* on chromosome I and at *leu1$^+$* on chromosome II. Red bars, two centromeres 1 and 2. (**D**) Western blot analyses of Slp1$^{Cdc20}$ protein levels in metaphase-arrested cells. Slp1$^{Cdc20}$ levels were normalized to those of total Cdc2 for each sample, with the relative ratio between Slp1$^{Cdc20}$ and Cdc2 at 0 min in *nda3-KM311* without extra copies of *slp1$^+$* set as 1.0. Note that in the strain carrying both endogenous *slp1$^+$* and *lys1Δ:: slp1$^+$*, Slp1$^{Cdc20}$ level is close to 2.5 instead of 2 times of endogenous level, most likely due to its flanking *adh1* terminator sequence. Error bars correspond to standard deviation. *p* values were calculated against the strain of *nda3-KM311* without extra copies of *slp1$^+$*. ***, *p*<0.001; **, *p*<0.01; n.s., not significant. (**E**) Elevated Slp1$^{Cdc20}$ levels rescue the TBZ sensitivity of *dnt1Δ* cells. (**F**) Elevated Slp1$^{Cdc20}$ levels reverse the anaphase initiation defect upon SAC inactivation in *dnt1Δ* cells. (**G**) Schematic summary of data from (**B-F**). The balance of relative abundance between Slp1$^{Cdc20}$ and checkpoint proteins governs the robustness of spindle checkpoint and thus mitotic exit rate. Lighter or darker shading depicts lower or higher levels of Slp1$^{Cdc20}$ present in MCC or APC/C in *dnt1Δ* or *slp1$^+$*-overexpression (*slp1$^{oe}$*) cells.

Consequently, sensitivity of *dnt1Δ* cells to TBZ was largely but not completely suppressed by excessive Slp1$^{Cdc20}$ expression achieved by three copies of *slp1$^+$* (Fig 3E). It should be noted that although the ectopic expression of *leu1-32::slp1$^+$* restored Slp1$^{Cdc20}$ in *dnt1Δ* cells almost to the same levels as in wild-type cells, it still could not rescue sensitivity of *dnt1Δ* cells to TBZ (Fig 3E). Nevertheless, increased Slp1$^{Cdc20}$ abundance compromised the checkpoint activation in both wild-type and *dnt1Δ* cells, as revealed by lower percentages of cells with Cdc13-GFP at SPBs and metaphase-arrest upon cold-shock treatment, and the extent of failed checkpoint response was largely proportional to Slp1$^{Cdc20}$ levels (Fig 3F and 3G). This result is consistent with previous reports in both fission and budding yeast that have underlined the importance of accurate relative abundance between checkpoint proteins and Cdc20/Slp1$^{Cdc20}$, which sets an important determinant of checkpoint robustness [58, 59]. However, we noticed that the disappearance rate of Cdc13-GFP from SPBs was not accelerated in either wild-type or *dnt1Δ* background strains when Slp1$^{Cdc20}$ abundance was artificially increased, regardless of the copy number of Slp1$^{Cdc20}$ present (Fig 3F), indicating that the SAC silencing tempo was not altered by Slp1$^{Cdc20}$ overexpression.

Together, these results suggested that the excessive Slp1 by three copies of *slp1$^+$* antagonizes the negative effects of *dnt1* deletion on TBZ sensitivity and lowered Slp1$^{Cdc20}$ protein levels may only partly contribute to the TBZ sensitivity of *dnt1Δ* cells. The failure of complete rescue of the TBZ sensitivity of *dnt1Δ* cells by artificially increased Slp1$^{Cdc20}$ abundance is possibly due to the retained and prolonged MCC-APC/C association, which still poses inhibition on APC/C activity.

## Enhanced MCC-APC/C association and lowered Slp1$^{Cdc20}$ abundance in *dnt1Δ* cells can be reversed by depletion of Apc15

It has been shown in both human and fission yeast cells that Apc15 mediates MCC binding to APC/C and is required for Cdc20/Slp1 autoubiquitylation and its turnover by APC/C [22,55,60,61]. Since our above data suggested that the TBZ sensitivity of *dnt1Δ* cells is likely caused by at least two aspects of defects in this mutant: one is the prolonged MCC-APC/C association, and the other is the lowered Slp1$^{Cdc20}$ protein level, we reasoned that the absence of Apc15 may reverse the positive effect of deletion of *dnt1$^+$* on MCC-APC/C association and its negative effect on Slp1$^{Cdc20}$ levels.

We first investigated how the absence of *apc15$^+$* affected the MCC-APC/C interaction in *dnt1Δ* cells. By immunoprecipitation of Apc4/Lid1-TAP, we found that the absence of Dnt1 enhanced MCC-APC/C interaction as more Mad2 and Mad3 were co-immunoprecipitated in *dnt1Δ* cells when compared to those in wild-type cells, whereas the deletion of *apc15$^+$* abolished the MCC-APC/C interaction both in wild-type and *dnt1Δ* cells (Fig 4A). Quite unexpectedly, although the total cellular level of Slp1$^{Cdc20}$ was significantly attenuated in *dnt1Δ* cells compared to that in wild-type cells (see Input in Fig 4A and 4B), we observed much more Slp1$^{Cdc20}$ was co-immunoprecipitated by Apc4/Lid1-TAP but significantly reduced in

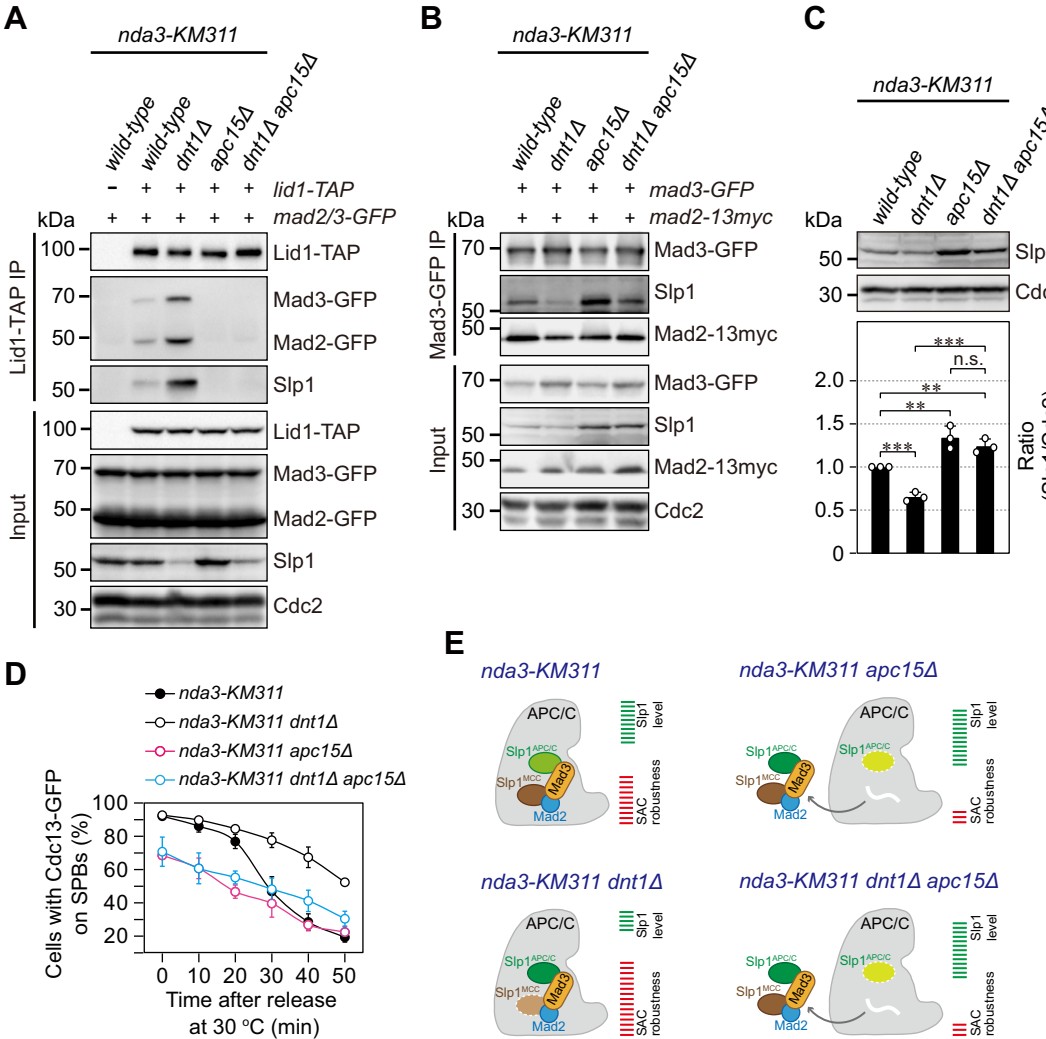

**Fig 4. *apc15Δ* reverses enhanced MCC-APC/C association and lowered Slp1^Cdc20^ abundance in *dnt1Δ* cells.** (A) *apc15Δ* abolishes the elevated binding of MCC to APC/C in *dnt1Δ* cells. The association of Mad2, Mad3 and Slp1^Cdc20^ to Apc4/Lid1 was assessed by immunoprecipitation of Apc4/Lid1-TAP in checkpoint-arrested cells as in Fig 2F. Note that more Mad2, Mad3 and Slp1^Cdc20^ was co-immunoprecipitated in *dnt1Δ* cells compared to wild-type cells, although the amount of Slp1^Cdc20^ was less abundant in *dnt1Δ* cells than that in wild-type cells. Results shown are the representative of three independent experiments. (B) *apc15Δ* restores the amount of Slp1^Cdc20^ bound to MCC in *dnt1Δ* cells. The assembly of MCC was assessed by immunoprecipitation of Mad3-GFP in checkpoint-arrested cells as in (A). Note that less Slp1^Cdc20^ was co-immunoprecipitated in *dnt1Δ* cells compared to wild-type cells. Results shown are the representative of three independent experiments. (C) *apc15Δ* restores the abundance of Slp1^Cdc20^ in *dnt1Δ* cells. Strains with indicated genotypes were grown and treated as in (A) to enrich checkpoint-arrested cells. Slp1^Cdc20^ levels were quantified with the relative ratio between Slp1^Cdc20^ and Cdc2 in wild-type strain set as 1.0. The experiments were repeated 3 times and the mean value for each sample was calculated as in Fig 3D. Error bars correspond to standard deviation. ***, $p<0.001$; **, $p<0.01$; n.s., not significant. (D) *apc15Δ* suppresses the delayed APC/C activation defect in *dnt1Δ* cells. Cells were synchronized by HU and then arrested at 18˚C for 6 hours before being released at 30˚C. The percentage of cells with Cdc13-GFP on SPBs was assessed at each time point as in Fig 2C. (E) Schematic summary of data from (A-D). White wavy lines depict the absence of Apc15. Lighter or darker shading depicts lower or higher levels of Slp1^Cdc20^ present in MCC or APC/C in mutant cells compared to wild-type cells.

Mad3-GFP immunocomplexes in *dnt1Δ* cells (Fig 4A and 4B). Interestingly, *apc15Δ* restored Slp1^Cdc20^ levels bound to MCC in *dnt1Δ* cells (Fig 4B). These observations seem to support the idea that the efficiency of Slp1^Cdc20^ being co-immunoprecipitated by APC/C subunit or MCC component differs significantly. Also, these data suggested that the increased APC/C-

associated Slp1$^{Cdc20}$ is likely "trapped" in the interface between MCC and APC/C and is unable to fulfill its function as the activator for APC/C.

Next, we examined whether depletion of Apc15 could rescue decreased Slp1$^{Cdc20}$ abundance in *dnt1Δ* cells. As expected, we indeed observed that the absence of *apc15$^+$* restored Slp1$^{Cdc20}$ to a level that was even higher than that in wild-type cells (Fig 4C). This quantitative data suggested that the attenuated Slp1$^{Cdc20}$ levels in *dnt1Δ* cells were most likely due to Apc15-faciltated degradation once Slp1$^{Cdc20}$ is incorporated in MCC.

Furthermore, the absence of Apc15 also compromised the spindle assembly checkpoint response to disruption of spindles in *dnt1Δ* cells due to the loss of MCC-APC/C association, as the percentages of cells with Cdc13-GFP at SPBs were sharply reduced from over 90% to only about 70% in *apc15Δ* cells after *nda3*-mediated SAC activation (Fig 4D). These results are consistent with previous report showing that the relatively higher amount of Slp1$^{Cdc20}$ can override the inhibitory effect of checkpoint proteins on APC/C activation [58]. Our results also suggested that Dnt1 is required for antagonizing the APC/C-mediated Slp1$^{Cdc20}$ degradation and maintaining Slp1$^{Cdc20}$ above its threshold level necessary for activating APC/C once the inhibitory signal from SAC is removed (Fig 4E).

## Human CUEDC2 can partially rescue the TBZ sensitivity and spindle checkpoint inactivation defects of *dnt1Δ* cells

In humans, CUEDC2 mediates the release of APC/C$^{Cdc20}$ activity from Mad2 inhibition, and depletion of CUEDC2 causes a checkpoint-dependent delay of the metaphase-anaphase transition [27]. Since deletion of *dnt1$^+$* in fission yeast also causes a checkpoint-dependent delay in anaphase entry, we wondered whether human CUEDC2 and fission yeast Dnt1 might share similar functions. To test this possibility, we ectopically expressed human CUEDC2 in fission yeast cells, and examined whether it could rescue the TBZ sensitivity and anaphase initiation defects upon SAC inactivation in *dnt1Δ* cells. Interestingly, the expression of nuclear localized CUEDC2 [i.e. CUEDC2 tagged with two copies of SV40 nuclear localization sequence (2×NLS)] slightly restored their growth at low concentrations of TBZ and efficiency of anaphase entry after SAC inactivation (S7 Fig). Cytoplasmic CUEDC2 did not have the same effect (S7B Fig), which was likely due to the fact that fission yeast undergoes "closed" mitosis, during which APC/C and MCC function only inside the nucleoplasm. On the contrary, similarly expressed human p31$^{comet}$, another factor with verified function in timely spindle checkpoint silencing by promoting the release of Mad2 from MCC [32–35, 37], failed to exert a rescuing effect on TBZ sensitivity of *dnt1Δ* cells (S7B Fig). Interestingly, we found limited homology between N-terminal portion of Dnt1 and CUEDC2 (S8 Fig). These data raised a possibility that fission yeast Dnt1 might be a functional homologue of human CUEDC2 and they may share a similar function in releasing APC/C$^{Cdc20}$ from checkpoint inhibition during mitotic exit.

## Dnt1 associates with APC/C upon SAC activation and during anaphase initiation

It has been previously shown that Cdk1-phosphorylated CUEDC2 directly binds to human Cdc20, and mediates the release of APC/C$^{Cdc20}$ activity from Mad2 inhibition [27]. To further dissect the possible mechanism that how Dnt1 promotes SAC inactivation, facilitates APC/C activation and maintains Slp1$^{Cdc20}$ stability, we examined whether Dnt1 also similarly associated with Slp1$^{Cdc20}$ or APC/C. By immunoprecipitation of sfGFP-Slp1 from SAC-activated cells arrested by *nda3-KM311*, we found limited amount of co-purified Dnt1 in addition to APC/C subunit Apc4/Lid1 and MCC component Mad2 (Fig 5A). Weak association between

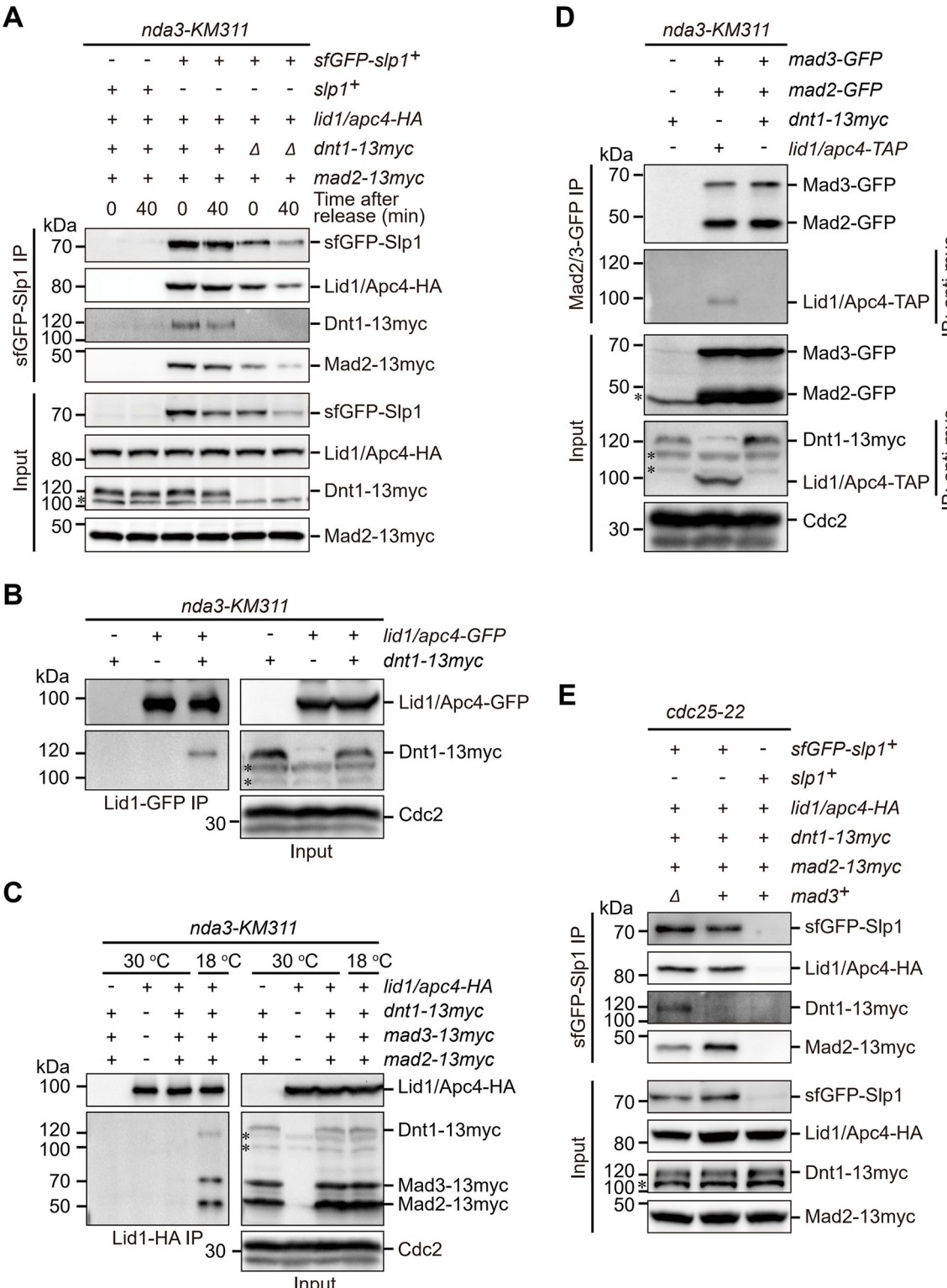

**Fig 5. Dnt1 associates with APC/C upon SAC activation and during anaphase initiation.** (**A**) Slp1$^{Cdc20}$ co-immunoprecipitates Dnt1 in metaphase-arrested and mitosis-exiting cells. *nda3-KM311 mad2-13myc* strains simultaneously carrying either *sfGFP-slp1* or *dnt1-13myc* or both were arrested in mitosis at 18°C and then released to 30°C, samples were collected at 0 and 40 min. Whole-cell extracts (input) were incubated with GFP-Trap beads and immunoprecipitated (IP) fractions were analyzed by immunoblotting. (**B, C**) Dnt1 can be co-immunoprecipitated by Apc4/Lid1 in metaphase-arrested but not asynchronous interphase cells. *nda3-KM311* strains

with indicated genotypes were either arrested at metaphase at 18˚C or grown as asynchronous cultures at 30˚C. Apc4/Lid1-GFP or Apc4/Lid1-HA was immunoprecipitated with GFP-Trap beads or anti-HA antibodies respectively and purified fractions were analyzed by immunoblotting. (**D**) Dnt1 cannot be co-immunoprecipitated by Mad2 and Mad3 in metaphase-arrested cells. *nda3-KM311* strains with indicated genotypes were arrested at metaphase as in (B). Both Mad2-GFP and Mad3-GFP were immunoprecipitated with GFP-Trap beads and purified fractions were analyzed by immunoblotting. The around 120 kDa band corresponding to Dnt1-13myc was not detectable in co-immunoprecipitated sample (lane #3). Note that one strain carrying *apc4/lid1-TAP* but without *dnt1-13myc* served as a positive control, and anti-myc antibodies cross-reacted with Apc4/Lid1-TAP. (**E**) Mad3 imposes inhibitory effect on association between Slp1 and Dnt1. *cdc25-22* strains expressing the indicated tagged proteins in the presence or absence of *mad3+* were first arrested at 36˚C for 3.5 hours and then released to 25˚C, samples were collected at 60 min after release. Immunoprecipitations were performed as in (A). In Fig 5A–5E, all asterisks indicate the unspecific band recognized by anti-myc antibodies, and all results shown are the representative of three independent experiments.

Dnt1 and APC/C was also detected in mitotically arrested but not asynchronized cultures by immunoprecipitation of Apc4/Lid1-TAP (Fig 5B and 5C). On the contrary, despite our intensive efforts, we were unable to detect the interaction between Mad2 or Mad3 and Dnt1 by co-immunoprecipitation (Fig 5D). These data suggested that very likely Dnt1 physically interacts more directly with APC/C but not MCC. Supporting this scenario, the interaction between sfGFP-Slp1 and Dnt1 was enhanced by depletion of Mad3 (Fig 5E), possibly because more APC/C assemblies devoid of MCC are available, thus Dnt1 gains better access to APC/C in *mad3Δ* cells. Unfortunately, we failed to detect the direct interaction between Dnt1 and APC/C subunits including Slp1$^{Cdc20}$ by yeast two-hybrid analyses (S6 Fig). These data suggested that possibly Dnt1 physically interacts with APC/C through a "mediating" factor, which remains to be identified in the future studies.

## Presence of Dnt1 is beneficial to maintaining APC/C activity and cell survival upon spindle checkpoint activation

Our above data suggested that the absence of Dnt1 causes prolonged and enhanced SAC activation. We wondered whether this defective SAC silencing is beneficial to cells. Very intriguingly, we noticed that individually isolated *nda3-KM311 dnt1Δ* cells formed far fewer colonies on solid rich medium at the permissive temperature 30˚C than *nda3-KM311 dnt1+* cells (47% vs. 68% viability) when they were treated at 18˚C for 10 hours to activate SAC (Fig 6A). In contrast, the viability of *dnt1Δ* cells was not affected by this transient cold-shock (Fig 6A). This data indicated that Dnt1 is required to maintain cell viability specifically upon spindle stress, although it is largely dispensable for spindle checkpoint activation (S2 Fig).

It has been previously shown that overexpression of spindle checkpoint protein Mad2 can activate SAC and block anaphase entry even in the absence of spindle defects [55,62], this is due to the fact that SAC proteins function as potent inhibitors of APC/C. Consistently, it is not surprising that fission yeast cells with temperature-sensitive mutations in the APC/C subunits were sensitive to elevated levels of Mad2 ($P_{nmt1}$-*mad2+*) [62] (Fig 6B). Given that Dnt1 is required to promote timely disassembly of MCC and cell survival, we examined whether elevated levels of Dnt1 can reverse the deleterious effects of overexpressed Mad2 on temperature-sensitive APC/C mutants. As expected, ectopic expression of *dnt1+* under the control of *adh1* promoter ($P_{adh1}$-*dnt1+*) rescued the growth defects of tested APC/C mutants overexpressing Mad2 (Fig 6B). Most strikingly, higher levels of Dnt1 allowed the survival of *cut20-100* $P_{nmt1}$-*mad2+* mutant, which is lethal by its own (Figs 6B and S9).

Together, these results revealed that Dnt1 is required for maintaining cell viability, and also for antagonizing SAC thus resulting timely and efficient anaphase initiation and mitotic exit, especially when the SAC needs to be inactivated.

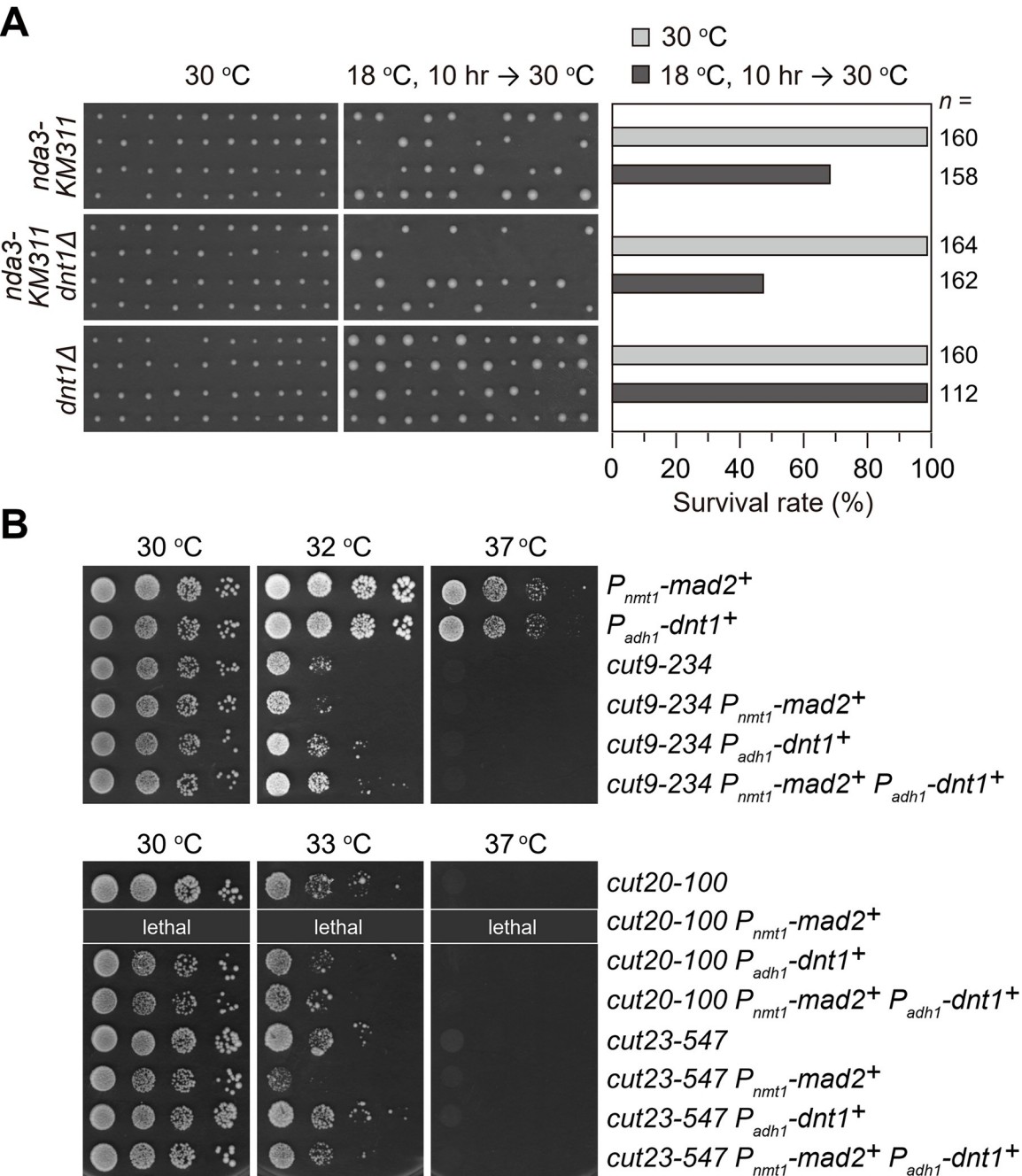

**Fig 6. Dnt1 involves in antagonizing SAC and maintaining cell survival upon spindle checkpoint activation.** (**A**) *nda3-KM311 dnt1Δ* cells lose viability after transient cold-shock. *nda3-KM311 dnt1⁺*, *nda3-KM311 dnt1Δ* or *dnt1Δ* cells were first grown at 30˚C in liquid cultures and then being shifted to 18˚C for 10 hours. Individual cells (*n* ≥80 for each strain) were isolated using tetrad dissection manipulator and placed on solid rich media at the permissive temperature of 30˚C (*Left*). The number of colonies formed before and after cold-treatment was quantified (*Right*). (**B**) Overexpressed Dnt1 antagonizes negative effect of Mad2 overexpression on loss-of-function APC/C mutants. Serial dilutions (10-fold) of the indicated strains were spotted on YE plates and incubated at the indicated temperatures. Overexpression of Mad2 and Dnt1 was achieved under *nmt1* promoter (*P_nmt1*) or *adh1* promoter (*P_adh1*), respectively. Note that *cut20-100 P_nmt1-mad2⁺* mutant was lethal and thus not included in the spot assay.

## Discussion

The anaphase-promoting complex/cyclosome (APC/C) is a large multisubunit ubiquitin ligase that triggers the metaphase-to-anaphase transition in the cell cycle by targeting the substrates cyclin B and securin for destruction. APC/C activity toward its substrates requires its co-activator Cdc20. To ensure that cells enter mitosis and partition their duplicated genome with high accuracy, APC/C$^{Cdc20}$ activity must be tightly controlled. So far, besides Cdc20, several factors have also been identified in higher eukaryotes to regulate APC/C$^{Cdc20}$ activity, such as mitotic protein kinases Cdk1 and Polo-like kinase 1 (Plk1), which increase Cdc20 binding and APC/C$^{Cdc20}$ activity by phosphorylating Apc1, Apc3 and likely also other subunits [63–65], and the mitotic checkpoint complex (MCC), which functions as a specific inhibitor of Cdc20 and APC/C$^{Cdc20}$ activity mediated mainly by Mad3/BubR1 binding to two molecules of Cdc20 [19,20]. The above Cdc20-regulating components and mechanisms are largely or most likely conserved in fission yeast [22,66].

In this study, we have identified the fission yeast nucleolar protein Dnt1 as a novel positive regulator of Slp1$^{Cdc20}$ protein level and APC/C$^{Cdc20}$ activity, especially when cells are recovered from arrest by activated spindle checkpoint. Although not tested, it is possible that our observed synthetic lethality between *dnt1Δ* and mutants with compromised kinetochores, peri-centromeric heterochromatin, cohesins, and microtubule-kinetochore attachment (S1 Fig) is due to low Slp1$^{Cdc20}$ level and resulted prolonged activated SAC signaling. Actually, one previous study has underlined the importance of accurate relative abundance both within checkpoint proteins and between checkpoint proteins and the checkpoint target Slp1$^{Cdc20}$ [58].

Our observations that deletion of *dnt1$^+$* causes elevated and prolonged MCC-APC/C association and reduced abundance of Slp1 when cells are recovered from activated spindle checkpoint arrest could be explained by several different possibilities. The simplest of these would be that Dnt1 directly interacts with Slp1$^{Cdc20}$ or other APC/C subunits to prevent the excessive and prolonged MCC binding to APC/C and consequently impede the degradation of MCC-bound Slp1$^{Cdc20}$ by partially activated APC/C, thus to maintain Slp1 above its threshold level when SAC is active (Fig 7). We favor the idea that Dnt1 directly regulates the MCC-APC/C, although the possibility of its indirect regulation cannot be excluded. In humans, depletion of CUEDC2 causes a checkpoint-dependent delay of the metaphase-anaphase transition, and Cdk1-phosphorylated CUEDC2 binds to Cdc20 directly and mediates the release of APC/C$^{Cdc20}$ activity from Mad2 inhibition [27]. Since deletion of *dnt1$^+$* in fission yeast also causes a checkpoint-dependent delay in anaphase entry and Dnt1 physically interacts with APC/C, it is thus plausible to assume that fission yeast Dnt1 might function similarly to human CUEDC2, though Dnt1 and CUEDC2 do not share high homology in their sequences (S8 Fig). However, we were unable to detect any interaction between Dnt1 and Slp1$^{Cdc20}$ and other APC/C subunits by yeast two-hybrid analysis (S6 Fig), this is very distinct from human CUEDC2, which directly binds to Cdc20 [27]. It is fairly possible that Dnt1 interacts with APC/C through an unidentified "mediator" protein, this may explain why Dnt1 comes down only weakly with Slp1$^{Cdc20}$ or Apc4/Lid1 in our immunoprecipitation assays (Fig 5). Therefore, we considered the fission yeast Dnt1 as a peripheral and phase-specific activator rather than an integral and permanent component or subunit of APC/C for two reasons. First, previous proteomics analyses of purified APC/C or potential Dnt1-interacting proteins did not identify Dnt1 or any APC/C subunits, respectively [44,67]. Second, we could detect the physical interaction between Dnt1 and Slp1$^{Cdc20}$ or Apc4/Lid1 by immunoprecipitation only in metaphase-arrested and mitosis-exiting cells (Fig 5A and 5B), but not in cells from asynchronized cultures (Fig 5C), suggesting Dnt1 may facilitate APC/C activity in a cell cycle stage-specific and transient manner. This timely involvement of Dnt1 may aid a rapid and complete release of APC/C activity once the SAC signaling is satisfied

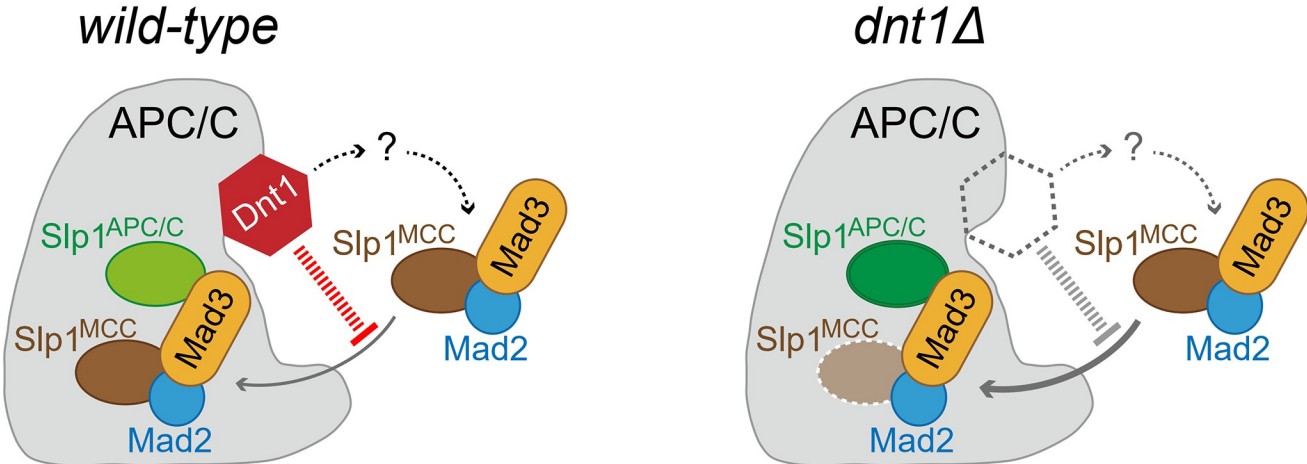

**Fig 7. Proposed model for positive regulation of the APC/C activity by Dnt1.** Upon SAC activation, APC/C-associated Dnt1 is likely involved in preventing the excessive and prolonged MCC binding to APC/C and thus impeding the degradation of MCC-bound Slp1$^{Cdc20}$ by partially activated APC/C. The presence of Dnt1 possibly help to maintain Slp1$^{Cdc20}$ protein (likely including both MCC-bound and free Slp1$^{Cdc20}$) level and APC/C$^{Cdc20}$ activity. Lighter brown color of Slp1$^{MCC}$ depicts lower levels of Slp1$^{Cdc20}$ present in MCC in *dnt1Δ* cells, which is likely caused by Apc15-faciliated autoubiquitylation and degradation. Darker green color of Slp1$^{APC/C}$ depicts higher levels of activation-incapable Slp1$^{Cdc20}$ "trapped" in APC/C in *dnt1Δ* cells. In parallel, Dnt1 may also indirectly regulate the inhibitory action of MCC on APC/C (dotted arrows) through a yet unrecognized factor/mechanism (question mark).

and switched off. It is possible that one or more extra unrecognized factors might exist in higher eukaryotes to regulate APC/C$^{Cdc20}$ activity in a way similar to Dnt1 does. In future, it would be interesting to know whether Dnt1 indeed interacts indirectly with Slp1$^{Cdc20}$ itself, and whether and how Dnt1 might antagonizes the autoubiquitylation of Slp1$^{Cdc20}$.

## Materials and methods

### Fission yeast strains and genetic analyses

Standard media (either YE (yeast extract) rich medium or EMM (Edinburgh minimal medium) and culturing methods were used [68,69]. G418 disulfate (Sigma-Aldrich), hygromycin B (Sangon Biotech) or nourseothricin (clonNAT; Werner BioAgents) was used at a final concentration of 100 μg/ml and thiabendazole (TBZ) (Sigma-Aldrich) at 5–15 μg/ml in YE media. For serial dilution spot assays, 10-fold dilutions of a mid-log-phase culture were plated on the indicated media and grown for 3 to 5 days at indicated temperatures. To examine the possible synthetic lethality of double mutants, at least 20 complete tetrads were dissected after each genetic cross. Yeast strains containing Cdc13-mCherry, Mad2-13myc and Mad3-13myc were generated by a PCR-based module method [70], with the DNA sequence information obtained from PomBase (https://www.pombase.org/). To create strains with an extra copy of *slp1*$^+$ or *dnt1*$^+$ at *lys1* locus, the *slp1*$^+$ genomic region from 1,504 bp 5' to stop codon of the open reading frame or 1852 bp of the open reading frame of *dnt1*$^+$ was first cloned into the vector pUC119-$P_{adh1}$-MCS-*hphMX6*-*lys1*$^*$ with *adh21* promoter ($P_{adh21}$) maintained or removed using the 'T-type' enzyme-free cloning method [71]. Similar procedures were employed to construct pUC119-$P_{adh21}$-MCS-*hphMX6*-*lys1*$^*$-based plasmids carrying sequences corresponding to human p31$^{comet}$ or CUEDC2 (with plasmid or cDNA kindly provided by Hongtao Yu or Jiahuai Han respectively). Then, GFP or two tandem SV40 NLS or both sequences was introduced in front of *dnt1*$^+$, p31$^{comet}$ or CUEDC2 coding sequences by Quikgene method [72]. Finally, the resultant plasmids were linearized by *Apa*I and integrated into the *lys1* locus, generating the strains *lys1Δ::P_{slp1}-slp1$^+$-T_{slp1}::hphMX6, lys1Δ::P_{slp1}-slp1$^+$-T_{adh1}::hphMX6, lys1Δ::P_{adh21}-dnt1$^+$-T_{adh1}::hphMX6,*

$lys1\Delta$::$P_{adh21}$-GFP-$dnt1^+$-$T_{adh1}$::hphMX6, $lys1\Delta$::$P_{adh21}$-GFP-$p31^{comet}$-$T_{adh1}$::hphMX6, $lys1\Delta$::$P_{adh21}$-2×NLS-$p31^{comet}$-$T_{adh1}$::hphMX6, $lys1\Delta$::$P_{adh21}$-2×NLS-GFP-$p31^{comet}$-$T_{adh1}$::hphMX6, $lys1\Delta$::$P_{adh21}$-GFP-CUEDC2-$T_{adh1}$::hphMX6, $lys1\Delta$::$P_{adh21}$-2×NLS-CUEDC2-$T_{adh1}$::hphMX6, and $lys1\Delta$::$P_{adh21}$-2×NLS-GFP-CUEDC2-$T_{adh1}$::hphMX6. The other strain with an extra copy of $slp1^+$ at $leu1$ locus has been described previously [58]. Strains used in this study are listed in S1 Table.

### Minichromosome loss assay

The minichromosome loss assay was performed in cells bearing the Ch16 ($ade6$-M216) minichromosome and $ade6$-M210 allele as previously described [54,73]. The rate of minichromosome loss per division was calculated by dividing the number of at least half red-sectored colonies grown on YE containing 12 mg/L adenine at 30°C by the number of total colonies (red colonies were excluded from the count).

### Cell synchronization methods

For $cdc25$-22 strains, cells were grown at 25°C until mid-log phase and arrested at late G2 phase by shifting to 36°C for 3.5 hr and released at 25°C. For $nda3$-KM311 strains, cells were grown at 30°C to mid-log phase, synchronized at S phase by adding HU (Sangon Biotech) to a final concentration of 12 mM for 2 hr followed by a second dose of HU (6 mM final concentration) for 3.5 hr. HU was then washed out and cells were released at specific temperatures as required by subsequent experiments.

### Spindle checkpoint activation assay

For metaphase arrest due to spindle checkpoint activation by disrupting mitotic spindles and abolishing kinetochore-microtubule attachment using the cold-sensitive $nda3$-KM311 mutation, cells synchronized by HU at 30°C were released to 18°C for up to 9 hr. For spindle checkpoint activation by the absence of tension generated by a temperature-sensitive mutation of a cohesin subunit, $psc3$-1T, cells were first arrested at S phase by HU at 25°C and then released to 37°C. For both methods, cells were withdrawn at certain time intervals and fixed with cold methanol and stained with DAPI (4', 6-diamidino-2-phenylindole, Sigma-Aldrich). 200–300 cells were analyzed for each time point.

### Spindle checkpoint silencing assay

For checkpoint silencing assay in the absence of microtubules, mid-log phase $ark1$-as3 $cdc13$-GFP $nda3$-KM311 cultures were first synchronized with HU at 30°C and then arrested in early mitosis by shifting to 18°C for 6 hr. 5μM ATP analog 1-NM-PP1 (Toronto Research Chemicals) was added to inactivate $ark1$-as3 and therefore spindle checkpoint. For checkpoint inactivation by shifting microtubule-depolymerized $nda3$-KM311 cells back to permissive temperature, cells were first synchronized with HU at 30°C and then arrested in early mitosis by shifting to 18°C for 6 hr, followed by incubation at 30°C to allow spindle reformation and therefore spindle checkpoint silencing. For both methods, cells were withdrawn at certain time intervals and fixed with cold methanol and stained with DAPI. 200–300 cells were analyzed for each time point. Each experiment was repeated at least three times.

### Yeast two-hybrid assay

For yeast two-hybrid analysis, the Matchmaker system (Clontech) was used. Bait plasmids were constructed into pGBKT7 vector. Prey plasmids were constructed into pGADT7 vector. Bait and prey plasmids were co-transformed into the AH109 strain, and transformants were

selected on double dropout medium (SD-Leu-Trp). The bait-prey interaction, which would activate the *HIS3* and *ADE2* reporter genes, was assessed by the growth on the triple (SD-Leu-Trp-His) or quadruple (SD-Leu-Trp-Ade-His) dropout media.

## Immunoblotting, immunoprecipitation and antibodies

Western blot and immunoprecipitation experiments were performed essentially as previously described [44]. Proteins were immunoprecipitated by IgG Sepharose beads (GE Healthcare) (for Apc4/Lid1-TAP), GFP-Trap beads (ChromoTek) (for sfGFP-Slp1 and Apc4/Lid1-GFP) or anti-HA antibody-coupled protein A/G beads (for Apc4/Lid1-HA). When necessary, 300 mM instead of 150 mM NaCl was used in lysate buffer to remove unspecific binding of proteins to beads. The primary antibodies used for immunoblot analysis of cell lysates and immunoprecipitates were peroxidase-anti-peroxidase (PAP) soluble complex (Sigma-Aldrich), rabbit polyclonal anti-Myc (GeneScript), mouse monoclonal anti-GFP (Beijing Ray Antibody Biotech), rat monoclonal anti-HA (Roche), mouse monoclonal anti-Cdc13 antibodies (Novus Biologicals) or rabbit polyclonal anti-Slp1 (generated at Xiamen University antibody facility using recombinant N-terminal 290 amino acids region of Slp1 (6His-Slp1(1-290aa) as antigens, same as previously described [57]). Cdc2 was detected using rabbit polyclonal anti-PSTAIRE (sc-53, Santa Cruz Biotechnology) as loading controls. Secondary antibodies were anti-mouse or anti-rabbit HRP conjugates (Thermo Fisher Scientific) and were read out using chemiluminescence.

## Fluorescence microscopy and live-cell imaging

GFP- and mCherry-fusion proteins (such as Cdc13-GFP, Cut2-GFP, Dnt1-GFP, CUEDC2-GFP, Cdc13-mCherry, and mCherry-Atb2) were observed in cells after fixation with cold methanol. Cells were washed in PBS and resuspended in PBS plus 1 µg/ml DAPI. Photomicrographs were obtained using a Nikon 80i fluorescence microscope coupled to a cooled CCD camera (Hamamatsu, ORCA-ER). Time-lapse imaging of live cells was performed at 30˚C using a Perkin Elmer spinning-disk confocal microscope (UltraVIEW VoX) with a 100x NA 1.49 TIRF oil immersion objective (Nikon) coupled to a cooled CCD camera (9100–50 EMCCD; Hamamatsu Photonics) and spinning disk head (CSU-X1, Yokogawa). Image processing, analysis and spindle length measurement were carried out using Element software (Nikon), ImageJ software (National Institutes of Health) and Adobe Photoshop.

## Statistical analysis

For quantitative analyses of each experiment, at least 200 cells were counted for each time point or sample, and each experiment was conducted at least three times. Error bars correspond to standard deviation (SD) throughout.

## Supporting information

**S1 Fig. Summary of synthetic lethality between *dnt1Δ* and mutants with defective chromosome segregation.**
(PDF)

**S2 Fig. Dnt1 is dispensable for activating the SAC in the absence of kinetochore-microtubule attachment or tension.**
(PDF)

**S3 Fig. Dnt1 is required to efficiently silence the spindle checkpoint when Aurora B kinase is inhibited in the absence of microtubules.**
(PDF)

**S4 Fig. Dnt1 is required for timely degradation of securin (Cut2 in *S. pombe*) upon SAC inactivation.**
(PDF)

**S5 Fig. Dma1 does not affect the abundance of Slp1$^{Cdc20}$ or restore Slp1$^{Cdc20}$ protein level in *dnt1Δ* cells during anaphase after SAC inactivation.**
(PDF)

**S6 Fig. Dnt1 does not directly associate with APC/C subunits based on yeast two-hybrid assays.**
(PDF)

**S7 Fig. Human CUEDC2 partially rescues the TBZ sensitivity and spindle checkpoint inactivation defects of *dnt1Δ* cells.**
(PDF)

**S8 Fig. Sequence alignment of *S. pombe* Dnt1 and human CUEDC2.**
(PDF)

**S9 Fig. Elevated expression of Mad2 causes strong synthetic lethality in *cdc20-100* mutant background.**
(PDF)

**S10 Fig. Original images of uncropped blots.**
(PDF)

**S1 Table. Yeast strains used in this study.**
(DOC)

**S2 Table. Raw numerical data.**
(XLSX)

## Acknowledgments

We thank Drs. Kathy Gould, Ursula Fleig, Xiangwei He, Iain Hagan, Silke Hauf, Jonathan Millar, Kevin G. Hardwick, Juraj Gregan, Takashi Toda, Yoshinori Watanabe, Mitsuhiro Yanagida, Hongtao Yu, Jiahuai Han and National BioResource Project, Japan (http://yeast.nig.ac.jp/yeast/) for fission yeast strains or plasmids.

## Author Contributions

**Conceptualization:** Quan-wen Jin.

**Data curation:** Quan-wen Jin.

**Formal analysis:** Shuang Bai, Li Sun, Quan-wen Jin.

**Funding acquisition:** Quan-wen Jin.

**Investigation:** Shuang Bai, Li Sun, Xi Wang, Shuang-min Wang, Zhou-qing Luo, Yamei Wang, Quan-wen Jin.

**Methodology:** Yamei Wang, Quan-wen Jin.

**Project administration:** Quan-wen Jin.

**Resources:** Quan-wen Jin.

**Supervision:** Yamei Wang, Quan-wen Jin.

**Validation:** Li Sun, Quan-wen Jin.

**Visualization:** Shuang Bai, Li Sun, Xi Wang, Quan-wen Jin.

**Writing – original draft:** Quan-wen Jin.

**Writing – review & editing:** Zhou-qing Luo, Yamei Wang, Quan-wen Jin.

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
