## [Decision Letter · Decision Letter 0]

22 Jun 2022

Dear Dr Jin,

Thank you very much for submitting your Research Article entitled 'Recovery from spindle checkpoint-mediated arrest requires a novel Dnt1-dependent APC/C activation mechanism' to PLOS Genetics.

The manuscript was fully evaluated at the editorial level and by two independent peer reviewers. The reviewers appreciated the attention to an important problem, but raised some substantial concerns about the current manuscript. Based on the reviews, we will not be able to accept this version of the manuscript, but we would be willing to review a revised version. We cannot, of course, promise publication at that time.

If you decide to revise the manuscript for further consideration at PLOS Genetics, please aim to resubmit within the next 60 days, unless it will take extra time to address the concerns of the reviewers, in which case we would appreciate an expected resubmission date by email to plosgenetics@plos.org.

[LINK]

We are sorry that we cannot be more positive about your manuscript at this stage. Please do not hesitate to contact us if you have any concerns or questions.

Yours sincerely,

Gregory P. Copenhaver

Editor-in-Chief

PLOS Genetics

Reviewer's Responses to Questions

**Comments to the Authors:**

Reviewer #1: This manuscript describes interesting observations showing that loss of function of the nucleolar protein Dnt1 causes a delay in anaphase inactivation of the mitotic checkpoint complex (MCC) and a consequent delay in activation of the anaphase promoting complex (APC). These effects seem to be mediated through the APC co-activator Slp1 (aka Cdc20 in other systems). The authors show reduced Slp1 levels and persistent anaphase association of Slp1 and the MCC with the APC in dnt1∆ mutant cells. While these results are interesting and in general convincing, the underlying mechanism remains unclear. The impact of the manuscript would be greatly enhanced if the authors could clearly show mechanistically whether Dnt1 is directly or indirectly involved in the phenomena described and what is the underlying mechanism.

In addition, the manuscript is confusing and difficult to follow what the authors think their experimental results mean as you read them. It would be very helpful to build their model in the results as the manuscript progresses. For example, they should describe clearly the rationale for experiments, what hypotheses are being tested and whether the results are consistent with the hypothesis. As is, it is not clear always clear how the results fit into an overall model for what is going on.

Main points.

1) The authors want to make the case that Dnt1 directly regulates the MCC-APC. However, Dnt1 is already proposed to have a numerous other functions as well as being a component of the nucleolus. I wonder if the effects they are seeing are indirect and represent a downstream consequence to other defects in these cells.

The main argument for a direct role for Dnt1 is in Figure 5 which seeks to show that Dnt1 associates with Slp1 specifically in metaphase and anaphase and not asynchronous cells. These experiments seem a bit preliminary and should be taken further. The association of Dnt1 with Slp1 is shown in checkpoint arrested (metaphase) nda3 (tubulin mutant) cells and in these cells 40 minutes after release from the arrest when they are in anaphase. The data from asynchronous cells is not shown and should be. The other way the Dnt1-Slp1 association is shown is in mitotic cells generated through release from a G2 arrest (cdc25-22). However the association is not observed in these cells unless mad3 is deleted. It is not clear why mad3∆ would increase the interaction of Dnt1 with Slp1. The authors state that their data suggests that “Dnt1 physically interacts more directly with APC/C but not MCC. Supporting this scenario, the interaction between sfGFP-Slp1 and Dnt1 was enhanced by depletion of Mad3 (Fig. 5B).” I do not understand why their data suggests this, since Mad3 associates with Slp1 both when it is bound to APC and when it is not bound to APC. Why not test whether Dnt1 comes down with the APC directly by Lid1/Apc4 IPs in mitosis. This would bolster the argument that it is associating with the APC and not the free MCC.

Extensive proteomics previous proteomics analysis on the APC has not identified Dnt1, so the observed association with Dnt1 is surprising. Also, why does Dnt1 not come down with Mad3 or Mad2, which are associated with Slp1 both on and off the APC. I wonder whether Dnt1 comes down weakly with Slp1 because it is a substrate.

2) I had a hard time understanding the logic and overall conclusions in the section titled: “Enhanced MCC-APC/C association and lowered Slp1Cdc20 abundance in dnt1∆ cells can be reversed by depletion of Apc15”. The overall conclusion is well stated at the end of the section. It would be helpful to state this at the beginning by saying something like: “One possibility for Dnt1 function is…… Several lines of evidence support this model…” This would give the reader some context to make the rationale and significance of the subsequent experiments more apparent. This same critique (and suggestion) could be applied for much of the manuscript.

3) In Figure 4C the authors state that they are looking at the amount of Slp1 associated with the MCC but not the APC by examining how much Slp1 comes down in Mad3 immunoprecipitations. But if you IP Mad3 you should get both pools of Mad3 associated with just the MCC and pools associated with the APC, so I do not understand the rationale for why Mad3 IPs only pull down Slp1 associated with the MCC and not the APC.

Minor Points.

1) Lines 217-219: States “Consistently, deletions of mad2+ or bub1+ completely abrogated this effect of dnt1∆ on delaying spindle checkpoint inactivation (Fig. 2I).” I do not think this makes sense because in these cells there is no spindle checkpoint in the first place, so you cannot look at the effect dnt1∆ on checkpoint inactivation. I think these sentences could be deleted.

2) Lines 280-282: “……whereas the deletion of apc15+ abolished the MCC-APC/C interaction both in wild-type and dnt1∆ cells (Fig. 4B).” This seems to be true for Mad2 and Slp1, but not Mad3. Is this significant and has it been reported previously?

3) In Figure 6, why do the authors think that dnt1∆ cells do not survive prolonged checkpoint arrest and release? If anything one might think that they would be better at surviving because they arrest better. It seems more likely that they do not decisively turn off the checkpoint and activate the APC in anaphase, leaving some securin and cyclin around and that interferes with anaphase chromosome segregation and mitotic exit.

Reviewer #2: In this manuscript the author has examined the role of dnt1 in the timely inactivation of spindle assembly checkpoint to efficiently initiate anaphase. These events are crucial for proper recovery of cells after APC mediated inhibition by spindle checkpoint proteins and mitotic exit. The author have shown that the Dnt1 inhibit prolonged association of mitotic checkpoint complex with APC/C during spindle assembly checkpoint inactivation. The genetic interaction studies and expression analysis revealed that the Dnt1 positively regulates the protein level of Slp1/Cdc20 protein.

Over all the manuscript is well written and observations are informative and very interesting, there are some clarifications that I would like to be addressed before publication.

Fig. 2F and G. The association of Mad2 with Lid1 in the absence of dnt1 persist longer than Mad3 association (Fig. F) but in graph (G) it is shown otherwise.

Fig. 2A why there are two different concentration of HU was used for synchronization?

Fig. 3A The negative genetic interaction was observed with APC subunits cut9, cut20 and cut23 mutants but not with nuc2 and apc15 mutants. Author could please explain the reason for the same.

Fig. 3A Why slp1-362 mutant is not growing properly even at 25oC,

Fig. 3B The nda3KM311 synchronization strategy was not properly mentioned in figure legend like how long the cells were incubated at low temperature before releasing them at 30oC.

Fig. 3D If author has tried to express the Slp1 under nmt1 promoter to suppress the TBZ sensitivity of dnt1Δ cells. Further what could be the reason slp1 expression at leu1 locus cannot suppress the TBZ sensitivity but expression at lys1 locus suppresses well.

**Have all data underlying the figures and results presented in the manuscript been provided?**

Reviewer #1: **No: **Underlying numerical data not provided.

Reviewer #2: Yes

PLOS authors have the option to publish the peer review history of their article (what does this mean?). If published, this will include your full peer review and any attached files.

Reviewer #1: No

Reviewer #2: No

---

## [Decision Letter · Decision Letter 1]

24 Aug 2022

Dear Dr Jin,

We are pleased to inform you that your manuscript entitled "Recovery from spindle checkpoint-mediated arrest requires a novel Dnt1-dependent APC/C activation mechanism" has been editorially accepted for publication in PLOS Genetics. Congratulations!

Yours sincerely,

Gregory P. Copenhaver

Editor-in-Chief

PLOS Genetics

Comments from the reviewers (if applicable):

Reviewer's Responses to Questions

**Comments to the Authors:**

Reviewer #1: The authors have fully addressed my concerns.

Reviewer #2: The authors have addressed all of the specific comments raised by this reviewer satisfactorily. The revised manuscript is substantially improved and this reviewer strongly recommends the manuscript to be accepted.

**Have all data underlying the figures and results presented in the manuscript been provided?**

Reviewer #1: Yes

Reviewer #2: Yes

PLOS authors have the option to publish the peer review history of their article (what does this mean?). If published, this will include your full peer review and any attached files.

Reviewer #1: No

Reviewer #2: No

**Data Deposition**

http://datadryad.org/submit?journalID=pgenetics&manu=PGENETICS-D-22-00538R1

**Press Queries**

---

## [Editor Report · Acceptance letter]

9 Sep 2022

PGENETICS-D-22-00538R1 

Recovery from spindle checkpoint-mediated arrest requires a novel Dnt1-dependent APC/C activation mechanism 

Dear Dr Jin, 

We are pleased to inform you that your manuscript entitled "Recovery from spindle checkpoint-mediated arrest requires a novel Dnt1-dependent APC/C activation mechanism" has been formally accepted for publication in PLOS Genetics! Your manuscript is now with our production department and you will be notified of the publication date in due course.

With kind regards,

Anita Estes

PLOS Genetics

On behalf of:
